# DISENTANGLE AND ALIGN: STRUCTURED CONTRASTIVE LEARNING WITH SEMANTIC–DOMAIN SEPARATION

## ABSTRACT

Learning compact representations that preserve semantics while discarding nuisance variation is central to self-supervised learning (SSL). However, when training data come from heterogeneous domains, instance-level contrastive learning often treats cross-domain yet semantically similar samples as *false negatives* and entangles domain factors with semantic features, yielding domain-clustered representations that generalize poorly to novel domains. To address this issue, we propose **Structured Contrastive Learning (SCL)**. This unified framework jointly learns (i) a semantic representation $\mathbf{z}_s$ via semantic contrast, (ii) a domain representation $\mathbf{z}_d$ via domain contrast, and (iii) their disentanglement by minimizing the dependence (mutual information) between $\mathbf{z}_s$ and $\mathbf{z}_d$. This structure preserves domain-invariant semantics in $\mathbf{z}_s$ while isolating domain factors in $\mathbf{z}_d$, enabling robust self-supervised training on data from a mixture of domains and out-of-domain (OOD) generalization on novel domains. Theoretically, we proved that the training objective of SCL disentangles semantic ($\mathbf{z}_s$) and domain ($\mathbf{z}_d$) information, and minimizing their mutual information $I(\mathbf{z}_s; \mathbf{z}_d)$ can effectively improve model generalization ability under domain shift. Empirically, we evaluate SCL on multi-domain training and demonstrate strong generalization to novel domains across diverse datasets and modalities.

## 1 INTRODUCTION

Learning condensed representations from observations Mikolov et al. (2013) is a central theme in contemporary machine learning research. A representation that faithfully compresses semantics while filtering out nuisance variation supports a wide range of downstream tasks such as generation Brown et al. (2020), few-shot classification (He et al., 2016; Brown et al., 2020), retrieval Radford et al. (2021), and reasoning Wei et al. (2022). Self-supervised learning (SSL) is particularly appealing in this context: by eliminating reliance on human annotations, SSL scales naturally, and it can discover fundamental semantic structure with improved robustness and generalization ability since it does not rely on task-specific supervision.

Contrastive learning Chen et al. (2020a) is a widely studied SSL paradigm with demonstrated success across data modalities and domains (He et al., 2020; 2024; Hu et al., 2021; Zhang et al., 2022a; Wan et al., 2024). By bringing semantically similar representations closer and pushing dissimilar negatives apart, contrastive objectives can recover rich semantic information without human annotation. However, contrastive learning is highly sensitive to the selection of negative samples. Particularly, when training data come from heterogeneous domains (e.g., background, texture, device, style), domain differences can be easily confounded with semantic differences: *cross-domain yet semantically similar samples are often treated as false negatives*, pushing their representations apart and inflating cross-domain intra-class variance. As a result, standard contrastive pretraining has no explicit inductive bias to disentangle semantics from domain factors, so the learned representations retain substantial domain-specific signals. At test time on a novel domain, these domain-related components act as nuisance variables, obscuring underlying semantic structure and degrading Out-Of-Domain (OOD) generalization. To disentangle semantics from domain factors for multi-domain data, existing SSL approaches generally fall into three strands: (1) Label-free approaches, (2) Label-aware approaches, and (3) Adversarial disentanglement approaches. Label-free approaches aim to

identify the discrepancy between domain and semantic factors without leveraging domain labels Scalbert et al. (2023). A common heuristic simply assumes that high-frequency signals (e.g., edges, fine textures) encode semantics, while low-frequency signals (e.g., brightness, shading) encode domain/style, and thus enforces low-frequency alignment Scalbert et al. (2023) Yang & Soatto (2020) Xu et al. (2021) Yang et al. (2022b). While sensible in many vision settings, the approach may fail when semantics are expressed in low-frequency signals, such as determining weather conditions or road visibility, as enforcing low-frequency consistency removes essential information. Label-aware approaches first train domain-specific models, and then align semantics across domains via explicit cross-domain matching Kim et al. (2021b) Zhang et al. (2022b) Yang et al. (2022a). Such a scheme requires complex memory banks or pairwise matching schedules, which scale poorly as the number of domains grows, and still under-model the interplay between semantic and domain factors. Adversarial disentanglement approaches jointly train a domain discriminator and an adversarial semantic encoder Feng et al. (2019); Kalibhat et al. (2023); Ganin et al. (2016). The domain discriminator is trained to predict the domain label, and the encoder is trained to fool it. As discrete domain labels only coarsely approximate real-world domain variability, they fail to capture the nuanced similarities and distinctions across domains. Furthermore, adversarially "fooling" a discriminator does not enforce statistical independence between semantic and domain features and thus offers no guarantee of invariance. In practice, this reliance on in-domain labels without a deeper model of domain structure leaves methods brittle under open-world shift: when novel domains appear only at test time, models trained on source domains often fail to remove domain-specific factors from new samples. Taken together with the well-documented instability of adversarial training, these limitations motivate more systematic, principled approaches to disentangle semantics from domain cues.

Ideally, effective disentanglement requires an explicit characterization of what constitutes *domain information*. To tackle these issues, we propose Structured Contrastive Learning (SCL) that jointly learns semantic representation $\mathbf{z}_s$ and domain representation $\mathbf{z}_d$, while encouraging their disentanglement by minimizing the mutual information $I(\mathbf{z}_s; \mathbf{z}_d)$. It effectively prevents the interference between domain information and semantic information, enabling the extraction of purified semantic representations in noisy environments with mixed domains. In addition, such purified semantic representations can also generalize more effectively to unseen domains, since domain shift is primarily captured by $\mathbf{z}_d$. The main contribution of our work can be summarized as follows: (1) We propose SCL, which learn semantic representation $\mathbf{z}_s$ and domain representation $\mathbf{z}_d$ separately, and can disentangle semantic information and domain information systematically to improve self-supervised learning performance under various domains. (2) Theoretically, we show that the optimization objective of SCL provably disentangles semantic information and domain information. By analyzing the generalization ability of SCL under domain shift, we demonstrate that disentangling $\mathbf{z}_s$ and $\mathbf{z}_d$ can enhance the generalization ability of $\mathbf{z}_s$. (3) Empirically, we evaluate SCL on multi-domain benchmarks across various modalities with leave-one-domain-out protocols and show that it consistently outperforms state-of-the-art SSL and domain-generalization baselines, improving OOD generalization. The ablation studies further confirm the complementary roles of semantic contrast, domain contrast, and disentanglement.

## 2 RELATED WORK: ROBUST LEARNING WITHIN MULTIPLE DOMAINS

The problem of multi-domain learning refers to learning from datasets originating from multiple domains with varying distributions. The key challenge is the often substantial distribution shift across domains. To enable robust multi-domain learning and enhance model generalization ability, a common principle is to learn domain-invariant features Schoenauer-Sebag et al. (2019), so that a model trained on several source domains attains low prediction error on a previously unseen target domain. Prior work tackles this problem from three complementary angles: data, representations, and learning strategies Wang et al. (2022). Data-centric methods Peng et al. (2022); Volpi et al. (2018); Zhang et al. (2017a) expand the quantity and diversity of training samples through augmentation and generation. Representation-based approaches can be broadly categorized into two main lines. For domain-invariant representation methods, classic solutions include kernel-based methods Blanchard et al. (2021); Muandet et al. (2013), domain adversarial learning Gong et al. (2019); Jia et al. (2020), explicit feature alignment Li et al. (2018b); Peng et al. (2019), and invariant risk minimization Arjovsky et al. (2019); Zhang et al. (2021). For feature disentanglement methods, Lv et al. (2022) proposes a causality-inspired framework to separate causal from non-causal factors. Beyond causality-based approaches, generative modeling Wang et al. (2021); Ilse et al. (2020);

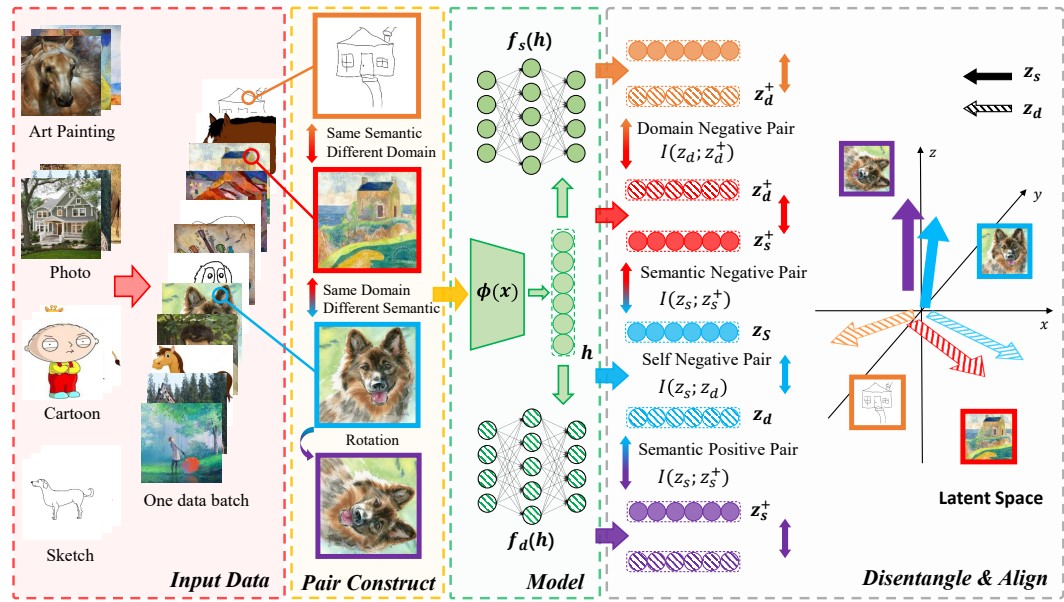

Figure 1: The overall Framework of SCL: Given input data from multiple domains, we aim to learn semantic representation $\mathbf{z}_s$ and domain representation $\mathbf{z}_d$ for each sample. We align $\mathbf{z}_d$ for samples from the same domain, and align $\mathbf{z}_s$ for samples with shared semantics. Moreover, for the same sample, its $\mathbf{z}_s$ and $\mathbf{z}_d$ are trained to be disentangled.

Zhang et al. (2022b) is also commonly used to improve disentanglement for better generalization. In addition, a variety of learning strategies have been devised and applied to foster robust multi-domain learning. Influential works include meta-learning-based approaches Khoee et al. (2024); Finn et al. (2017); Li et al. (2018a), gradient-based approaches Huang et al. (2020), distributionally robust optimization-based approaches Zhai et al. (2021); Rahimian & Mehrotra (2019), and self-supervised learning-based approaches Kim et al. (2021a); Scalbert et al. (2023); Carlucci et al. (2019); Zhang et al. (2022b).

## 3    PROBLEM FORMULATION

A dataset collected from $k$ domains can be denoted as $\{(\mathbf{x}_i, y_i, d_i)\}_{i=1}^n \sim p(\mathbf{x}, y, d)$, where $\mathbf{x}_i$ is the $i$-th observed data point, $y_i$ is the semantic label associated with $\mathbf{x}_i$, and $d_i \in \{\mathcal{D}_1, \mathcal{D}_2, \ldots, \mathcal{D}_k\}$ is the domain label of sample $\mathbf{x}_i$, indicating its source or style. Our goal is to learn two proper mappings $f_1(\cdot)$ and $f_2(\cdot)$ that can extract $\mathbf{x}$'s semantics information and domain information properly as $\mathbf{z}_s = f_1(\mathbf{x})$ and $\mathbf{z}_d = f_2(\mathbf{x})$, where $\mathbf{z}_s$ is the semantic representation, which can be utilized for various downstream tasks involving semantic information, such as predicting semantic label $y$; $\mathbf{z}_d$ is the domain representation, involving domain information and can be learned through domain-related signals. In our self-supervised learning setting, we can only access $\{\mathbf{x}_i, d_i\}_{i=1}^n$ in the training phase and the semantic label $\{y_i\}_{i=1}^n$ is not accessible.

## 4    METHODOLOGY

### 4.1    MOTIVATION

Multi-domain observations are generated by three latent factors: semantics $\mathbf{S}$, domain/style $\mathbf{D}$, and nuisance noise $\epsilon$, through a mechanism $\mathbf{x} = h(\mathbf{S}, \mathbf{D}, \epsilon)$. We build an encoder $g(\cdot)$ with two heads $f_s(\cdot), f_d(\cdot)$ producing $\mathbf{z}_s = f_s(g(\mathbf{x})), \mathbf{z}_d = f_d(g(\mathbf{x}))$ aiming for (i) $\mathbf{z}_s$ to capture domain-invariant semantics, (ii) $\mathbf{z}_d$ to capture domain/style information, and (iii) the two to be disentangled. The ideal goal can be summarized as: $I(\mathbf{z}_s; \mathbf{S}) = H(\mathbf{S})$, $I(\mathbf{z}_d; \mathbf{D}) = H(\mathbf{D})$, and $I(\mathbf{z}_s; \mathbf{z}_d) = 0$, where $I(\cdot; \cdot)$ is the mutual information between two random variables, and $H(\cdot)$ denotes information entropy

Shannon (1948). We can achieve this objective by minimizing the following loss function:

$$\mathcal{L} = -I(\mathbf{z}_s; \mathbf{S}) - \lambda_1 I(\mathbf{z}_d; \mathbf{D}) + \lambda_2 I(\mathbf{z}_s; \mathbf{z}_d), \tag{1}$$

where $\lambda_1$ and $\lambda_2$ are hyperparameters adjusting values for each term. Since $\mathbf{S}$ and $\mathbf{D}$ are not directly accessible in our case, we optimize the following term instead:

$$\mathcal{L}^{\star} = -I(\mathbf{z}_s; \mathbf{z}_s^+) - \lambda_1 I(\mathbf{z}_d; \mathbf{z}_d^+) + \lambda_2 I(\mathbf{z}_s; \mathbf{z}_d), \tag{2}$$

where $I(\mathbf{z}_s; \mathbf{z}_s^+)$ denotes the mutual information between semantic representations for sample pairs that share the same semantics. Since semantic labels are not available during training, we construct semantic positive pairs by augmenting the same sample. $I(\mathbf{z}_d; \mathbf{z}_d^+)$ denotes the mutual information between domain representations for sample pairs that come from the same domain; and $I(\mathbf{z}_s; \mathbf{z}_d)$ denotes the mutual information between the same sample's semantic representation and its domain representation. Optimizing this objective enables us to align samples with the same semantics in the semantic representation space, align samples from the same domain in the domain representation space, and simultaneously disentangle semantic and domain representations. We will prove that optimizing Equation (1) and Equation (2) are equivalent under certain conditions in the Theoretical Analysis section. In practice, we optimize InfoNCE loss as a proxy for $I(\mathbf{z}_s; \mathbf{z}_s^+)$ and $I(\mathbf{z}_d; \mathbf{z}_d^+)$ Oord et al. (2018), and optimize Hilbert–Schmidt Independence Criterion (HSIC) as a proxy for $I(\mathbf{z}_s; \mathbf{z}_d)$ Gretton et al. (2005).

## 4.2 MODEL STRUCTURE

For an input training dataset with domain labels denoted as $\{\mathbf{x}_i, d_i\}_{i=1}^n$, our proposed framework can be formulated in the following way. The backbone feature extractor $\phi(\cdot)$, typically parameterized as a Convolution Neural Network (CNN) for images, Multilayer Perceptron(MLPs) for tabular data, and Graph Neural Networks(GNNs) for Graph, is used to obtain shallow representations of the data $\mathbf{h}_i = \phi(\mathbf{x}_i)$, which contain entangled semantic and domain information. The extracted shallow representation $\mathbf{h}_i$ is then fed into two separate heads to obtain a semantic representation as $\mathbf{z}_{s,i} = f_s(\mathbf{h}_i)$, and a domain representation as $\mathbf{z}_{d,i} = f_d(\mathbf{h}_i)$. Both extractors $f_s(\cdot)$ and $f_d(\cdot)$ are parameterized as Multilayer Perceptrons(MLPs).

## 4.3 TRAINING

We train end-to-end by sampling a mini-batch $\mathcal{B}$ of size $B$, drawing two stochastic augmentations $t, t'$ per sample to form $\mathbf{x}_i^{(1)} = t(\mathbf{x}_i)$ and $\mathbf{x}_i^{(2)} = t'(\mathbf{x}_i)$, and computing $\mathbf{z}_{s,i} = f_s(\phi(\mathbf{x}_i^{(1)}))$, $\mathbf{z}_{s,i}^+ = f_s(\phi(\mathbf{x}_i^{(2)}))$ and $\mathbf{z}_{d,i} = f_d(\phi(\mathbf{x}_i^{(1)}))$, $\mathbf{z}_{d,i}^+ = f_d(\phi(\mathbf{x}_i^{(2)}))$. The semantic contrast uses instance-level positives and treats all other instances as negatives: for anchor $i$ the negative index set is $\mathcal{N}_s(i) = \{j \in \mathcal{B} \setminus \{i\}\}$, and the InfoNCE loss for this augmentation is defined as:

$$\ell_i^{(s)} = -\log \frac{\exp\{\mathrm{sim}(\mathbf{z}_{s,i}, \mathbf{z}_{s,i}^+)/\tau_s\}}{\exp\{\mathrm{sim}(\mathbf{z}_{s,i}, \mathbf{z}_{s,i}^+)/\tau_s\} + \sum_{j \in \mathcal{N}_s(i)} \exp\{\mathrm{sim}(\mathbf{z}_{s,i}, \mathbf{z}_{s,j})/\tau_s\}}, \tag{3}$$

where $\mathrm{sim}(\cdot, \cdot)$ denotes cosine similarity and $\tau_s$ is a temperature. The total InfoNCE loss for the mini-batch $\mathcal{B}$ is $\mathcal{L}_{\mathrm{sem}} = \frac{1}{B} \sum_{i \in \mathcal{B}} \ell_i^{(s)}$.

The domain contrast aligns within-domain samples and repels cross-domain ones by using domain labels to specify the positive and negative sets $\mathcal{P}_d(i) = \{j \in \mathcal{B} \setminus \{i\} : d_j = d_i\}$ and $\mathcal{N}_d(i) = \{j \in \mathcal{B} : d_j \neq d_i\}$. We adopt a multi-positive InfoNCE as:

$$\ell_i^{(d)} = -\log \frac{\sum_{j \in \mathcal{P}_d(i) \cup \{i\}} \exp\{\mathrm{sim}(\mathbf{z}_{d,i}, \mathbf{z}_{d,j})/\tau_d\}}{\sum_{j \in \mathcal{P}_d(i) \cup \{i\}} \exp\{\mathrm{sim}(\mathbf{z}_{d,i}, \mathbf{z}_{d,j})/\tau_d\} + \sum_{j \in \mathcal{N}_d(i)} \exp\{\mathrm{sim}(\mathbf{z}_{d,i}, \mathbf{z}_{d,j})/\tau_d\}}, \tag{4}$$

with temperature $\tau_d$. The aggregate domain loss for mini-batch $\mathcal{B}$ is $\mathcal{L}_{\mathrm{dom}} = \frac{1}{B} \sum_{i \in \mathcal{B}} \ell_i^{(d)}$.

To discourage information leakage between the two heads and approximate the minimization of $I(\mathbf{z}_s; \mathbf{z}_d)$, we use the Hilbert–Schmidt Independence Criterion (HSIC) Gretton et al. (2005) computed on the batch representations $\mathbf{Z}_s = [\mathbf{z}_{s,1}, \dots, \mathbf{z}_{s,B}]^\top$ and $\mathbf{Z}_d = [\mathbf{z}_{d,1}, \dots, \mathbf{z}_{d,B}]^\top$. Let $\mathbf{K}_{ij} = \exp(-\|\mathbf{z}_{s,i} - \mathbf{z}_{s,j}\|^2/2\sigma_s^2)$ and $\mathbf{L}_{ij} = \exp(-\|\mathbf{z}_{d,i} - \mathbf{z}_{d,j}\|^2/2\sigma_d^2)$ be Gaussian-kernel Gram

matrices with bandwidths set by the median heuristic, and $\mathbf{H} = \mathbf{I} - \frac{1}{B}\mathbf{1}\mathbf{1}^\top$ the centering matrix, the disentanglement penalty is:

$$\mathcal{L}_{\text{sep}} = \frac{1}{(B-1)^2}\operatorname{tr}(\mathbf{KHLH}), \qquad (5)$$

which equals zero if and only if $\mathbf{Z}_s$ and $\mathbf{Z}_d$ are independent for universal kernels Gretton et al. (2007). The overall objective is $\mathcal{L}_{\text{train}} = \mathcal{L}_{\text{sem}} + \lambda_1 \mathcal{L}_{\text{dom}} + \lambda_2 \mathcal{L}_{\text{sep}}$, and we jointly update $\phi(\cdot)$, $f_s(\cdot)$, and $f_d(\cdot)$ via stochastic gradient descent.

# 5 THEORETICAL ANALYSIS

## 5.1 DISCUSSION ON THE OPTIMIZATION OBJECTIVE

In this section, we will show that optimizing Equation (1) is equivalent to optimizing Equation (2) under certain conditions.

**Theorem 1.** *Let $\mathbf{S}$ be a latent semantic factor. Assume the two-view conditional independence $\mathbf{z}_s \perp \mathbf{z}_s^+ \mid \mathbf{S}$. Moreover, we assume that $\mathbf{z}_s$ and $\mathbf{z}_s^+$ are sufficient statistics for $\mathbf{S}$, then we have:*

$$\mathbf{argmax}_\theta I(\mathbf{z}_s; \mathbf{z}_s^+) = \mathbf{argmax}_\theta I(\mathbf{z}_s; \mathbf{S}), \qquad (6)$$

*where $\theta$ are parameters for encoder $f_s(\cdot)$, $f_d(\cdot)$ and $\phi(\cdot)$. Specific proof is in Appendix A.2.*

**Theorem 2.** *Let $d$ be a domain label. Assume the two-view conditional independence $\mathbf{z}_d \perp \mathbf{z}_d^+ \mid d$. Moreover, we assume that $\mathbf{z}_d$ and $\mathbf{z}_d^+$ are sufficient statistics for $d$, then we have:*

$$\mathbf{argmax}_\theta I(\mathbf{z}_d; \mathbf{z}_d^+) = \mathbf{argmax}_\theta I(\mathbf{z}_d; \mathbf{D}), \qquad (7)$$

*where $\theta$ are parameters for encoder $f_s(\cdot)$, $f_d(\cdot)$ and $\phi(\cdot)$. Specific proof is in Appendix A.2.*

Consequently, optimizing Equation (1) is equivalent to optimizing Equation (2) under the assumptions as mentioned above. The conditional independence assumptions often hold since each sample is stochastically transformed into two independently augmented views/observations, making the views conditionally independent given the underlying semantics/domain.

## 5.2 DISCUSSION ON GENERALIZATION

In this section, we consider a scenario involving only a source domain, $\mathcal{D}_1$, and a target domain, $\mathcal{D}_2$. Through theoretical analysis, we demonstrate that minimizing $I(\mathbf{z}_s; \mathbf{z}_d)$ improves generalization ability across domains. Consider a linear probe $h_w(\mathbf{z}_s) = \mathbf{w}^\top \mathbf{z}_s$ with $\|\mathbf{w}\| \le W$, which can decode the semantic representation $\mathbf{z}_s$ to predict label $\hat{y}$. Denote the true class label by $y$, The population risk on domain $d \in \{1, 2\}$ is defined by:

$$R_d(\mathbf{w}) = \mathbb{E}_{(\mathbf{z}_s, y) \sim \mathcal{D}_d}\big[\ell\big(y, \mathbf{w}^\top \mathbf{z}_s\big)\big], \qquad (8)$$

and the empirical risk on $\mathcal{D}_1$ is defined by $\hat{R}_1(\mathbf{w})$ with $n$ samples. We assume $\|\mathbf{z}_s\| \le B$ almost surely on both domains.

**Assumption 1.** *Either $p_1(y) = p_2(y)$ holds, or we evaluate $R_1$ and $\hat{R}_1$ under importance weighting so that the effective class prior matches $p_2(y)$.*

**Assumption 2.** *Let $\mathsf{k} : \mathbb{R}^p \times \mathbb{R}^p \to \mathbb{R}$ be a bounded, sufficiently smooth kernel (e.g., a Matérn kernel inducing a Sobolev RKHS of order $s > p/2 + 1$) on a compact support of $\mathbf{z}_s$. There exists a constant $C_\mathsf{k} > 0$ such that for any probability measures $P, Q$ on the support,*

$$W_1(P, Q) \le C_\mathsf{k} \operatorname{MMD}_\mathsf{k}(P, Q),$$

*where $W_1(\cdot, \cdot)$ is the Wasserstein distance and $\operatorname{MMD}_\mathsf{k}(\cdot, \cdot)$ is the Maximum Mean Discrepancy with kernel $\mathsf{k}$.*

**Assumption 3.** *Assume that, given a semantic label $y$, the domain signal is sufficiently captured by $\mathbf{z}_d$ so that the Markov chain $d \to \mathbf{z}_d \to \mathbf{z}_s$ holds conditionally on $y$.*

**Theorem 3** (Main bound; minimizing $I(\mathbf{z}_s; \mathbf{z}_d)$ improves target generalization). *Let $\hat{\mathbf{w}} = \mathbf{argmin}_{\|\mathbf{w}\| \leq W} \hat{R}_1(\mathbf{w})$ be the linear probe trained on $\mathcal{D}_1$'s semantic representation. Under Assumptions 1,2,3, with probability at least $1 - \delta$, we have:*

$$R_2(\hat{\mathbf{w}}) \leq \hat{R}_1(\hat{\mathbf{w}}) + 2\frac{WB}{\sqrt{n}} + 3\sqrt{\frac{\ln(2/\delta)}{2n}} + C\sqrt{I(\mathbf{z}_s; \mathbf{z}_d)}, \tag{9}$$

*where $C$ is a positive real number. Consequently, any training strategy that decreases $I(\mathbf{z}_s; \mathbf{z}_d)$ on $\mathcal{D}_1$ (while keeping $W, B$ controlled) provably tightens the upper bound on the target risk $R_2(\hat{\mathbf{w}})$ on $\mathcal{D}_2$. Specific proof is in Appendix A.2.*

Consequently, based on Equation (9), we theoretically prove that minimizing $I(\mathbf{z}_s, \mathbf{z}_d)$ can decrease $R_2(\hat{\mathbf{w}})$'s upper bound, and increase the generalization ability under domain shift. Intuitively speaking, domain shift occurs in the domain representation $\mathbf{z}_d$'s space, and minimizing $I(\mathbf{z}_s, \mathbf{z}_d)$ effectively strips away the influence of domain shift on semantic representation $\mathbf{z}_d$, thereby enhancing the generalization ability of $\mathbf{z}_s$ in new domains. Assumption 1 often holds, since domains sampled from the same dataset typically share similar class priors; Assumption 2 often holds, since we employ bounded, smooth kernels (e.g., Gaussian) on bounded representations; Assumption 3 often holds, since domain information is primarily captured by $\mathbf{z}_d$, with leakage into $\mathbf{z}_s$ suppressed during training.

# 6 EXPERIMENTS

## 6.1 DATASETS

We conducted experiments on three public image datasets and one private medical dataset. **MNIST-C** is a four-domain, synthetic dataset based on MNIST with four distinct "styles", : Domain 0 (Original) pre- serves the original appearance; Domain 1 (Solarized) applies exposure solarization with pronounced sharpening, denoted as "Original"; Domain 2 (Posterized) reduces tonal detail (posterization) and boosts overall contrast; Do- main 3 (Warped) introduces perspective warping together with a modest shear, creating geometric distortion while keeping the digit identity unchanged. **Rotated MNIST** is a four-domain dataset based on MNIST. Domains 1, 2, 3, and 4 correspond to images rotated clockwise by 0°, 30°, 60°, and 90°, respectively. **PACS** is a domain generalization benchmark with four visually distinct domains (Photo, Art Painting, Cartoon, Sketch) and seven object categories (dog, elephant, giraffe, guitar, horse, house, person). **ADNI** is a private tabular medical dataset collected from 39 sites, representing 39 domains. More details are shown in the appendix Appendix A.3.1.

## 6.2 BASELINE METHODS

To validate SCL's performance, we compare it with a diverse set of current methods towards Self-Supervised Learning and Domain Generalization. More details are shown in the appendix Appendix A.3.2.

- *Self-Supervised Learning Methods*: **SimCLR** Chen et al. (2020a) is one of the most widely used contrastive learning methods, and is well-known for its robustness in extracting semantics from redundant and noisy training samples without supervision. **Moco** He et al. (2020) is a well-known contrastive learning method that uses a momentum-updated encoder and a queue of negative samples. **Naive SCL** is a framework with the same structure as our proposed **SCL**. The only difference between **Naive SCL** and **SCL** is that we froze the optimization of the domain head and the disentanglement term when training **Naive SCL**, only optimizing the semantic head Chen et al. (2020a).

- *Domain Generalization Methods*: **Sagnet** Nam et al. (2021) is a domain generalization method that suppresses style cues—via feature-level style randomization and adversarial training—so models rely on content and generalize robustly to unseen domains. **SSRL-MD** Feng et al. (2019) trains a single self-supervised encoder on multi-domain data, combining a gradient-reversal domain classifier to remove domain cues with a contrastive Jensen–Shannon divergence (JSD) based term to preserve within-domain information.

Table 1: Performance comparison across four datasets, together with four domains in MNIST-C as the target domain. Each cell reports ID Accuracy and OOD Accuracy (%).

| Methods | MNIST-C | | PACS | | RotatedMNIST | | ADNI | | Original | | Solarized | | Posterized | | Warped | |
| --- | --- | --- | --- | --- | --- | --- | --- | --- | --- | --- | --- | --- | --- | --- | --- | --- |
| | ID | OOD | ID | OOD | ID | OOD | ID | OOD | ID | OOD | ID | OOD | ID | OOD | ID | OOD |
| SimCLR | 59.81 | 42.97 | 48.29 | 21.93 | 59.28 | 31.62 | 53.86 | 54.87 | 55.44 | 60.00 | 66.81 | 19.50 | 56.00 | 61.44 | 60.92 | 30.94 |
| MoCo | 27.34 | 22.65 | 46.92 | 21.67 | 61.62 | 34.43 | **61.93** | 51.28 | 24.49 | 25.69 | 26.53 | 20.80 | 26.22 | 29.72 | 32.13 | 14.40 |
| SagNet | 52.90 | 44.39 | 39.48 | 26.70 | 54.08 | 30.67 | 59.24 | 51.19 | 49.04 | 53.75 | 57.43 | **46.40** | 49.63 | 52.20 | 55.51 | 22.20 |
| SSRL-MD | 57.72 | 40.91 | **49.86** | **27.01** | 49.57 | 33.89 | 54.07 | 55.07 | 51.25 | 58.32 | 59.26 | 12.71 | 54.96 | 60.72 | 65.40 | 31.89 |
| DDM | 45.69 | 37.03 | 49.75 | 21.52 | 43.57 | 24.45 | 53.84 | 53.98 | 43.76 | 50.56 | 45.69 | 17.75 | 42.16 | 40.32 | 51.13 | **39.49** |
| Naive SCL | 61.31 | 45.03 | 48.97 | 22.07 | 60.71 | 33.73 | 54.01 | 55.26 | 57.09 | 63.22 | 65.89 | 17.96 | 59.01 | 63.69 | 64.08 | 35.49 |
| SCL (ours) | **63.47** | **47.62** | 49.05 | 24.80 | **65.06** | **37.29** | 54.31 | **55.80** | **59.29** | **65.82** | **66.65** | 18.21 | **61.33** | **67.06** | **64.17** | 37.47 |

**DDM** Kalibhat et al. (2023) adds a small domain-coded prefix and adversarially enforces domain invariance on the remaining features, with optional robust clustering for unlabeled domains—yielding stronger cross-domain SSL representations.

## 6.3 Evaluation Metrics and Backbone Setting

We employ a leave-one-domain-out protocol: our model is trained on the source domain and evaluated on both the source and target domains. The quality of $\mathbf{z}_s$ is assessed through its linear classification accuracy on semantic labels $y$. A linear probe trained on the source domain's representation yields In-Distribution (ID) Accuracy, and the same probe on the held-out target yields Out-of-Distribution (OOD) accuracy. The backbone model in image setting (MNIST-C, PACS, RotatedMNIST) is a lightweight CNN for single-channel inputs with three $3 \times 3$ convolutional blocks (channels $1 \rightarrow 32 \rightarrow 64 \rightarrow 128$) interleaved with pooling, followed by global average pooling and a linear layer $128 \rightarrow 256$; two MLP heads produce 128 dimension representations $\mathbf{z}_s$ and $\mathbf{z}_d$ from the 256 dimension feature. In the ADNI setting, we employ a two-layer MLP as the backbone model. The first linear layer projects the input features into a 128-dimensional space, followed by a second linear layer that further maps them into a 256-dimensional space. Analogous to the image-based setting, two additional MLP heads are applied to this 256-dimensional representation to produce the 128-dimensional representations $\mathbf{z}_s$ and $\mathbf{z}_d$. The semantic head uses a two-view "weak" augmentation: for each image, it independently samples two lightly perturbed views via gentle geometric jitter (e.g., small random crop with padding and slight rotation/translation), mild noise/blur, and standard normalization. These two views of the same instance form the positive pair for the InfoNCE objective, while all other instances in the batch serve as negatives. To evaluate the effectiveness of SCL, we report its performance together with that of the baseline methods on four datasets. For each dataset, every domain is sequentially designated as the target domain, with the remaining domains serving as source domains. The average performance is presented in Table 1. Furthermore, we provide results on the MNIST-C dataset in Table 1, where each of the four domains is considered as the target domain in turn, with the other domains regarded as source domains.

## 6.4 Numerical Performance

The experiment results in Table 1 show that, in comparison with various baseline methods, SCL consistently demonstrates superior performance in terms of both ID accuracy and OOD accuracy. This indicates that SCL not only achieves excellent performance in multi-domain training but also exhibits a strong cross-domain generalization ability compared to current methods. Further experiment results in Table 1 suggest that when the target domain is "Original", "Solarized", "Posterized", and "Warped", SCL demonstrates significantly superior performance in both ID accuracy and OOD accuracy.

## 6.5 Ablation Studies

To validate that our proposed SCL indeed contributes to *multiple domain self-supervised training* and *cross-domain generalization*, we compare it with Naive SCL. Notably, on the dataset MNIST-C, we executed train SCL and Naive SCL 10 times across three source domains, validating on the distinct target domain with each run. The average linear classification accuracy on source domains, as shown in Figure 2a, demonstrates that SCL consistently outperforms Naive SCL. The mean performance of SCL is 63.47%, which exceeded that of Naive SCL (61.31%) by approximately 2.16%

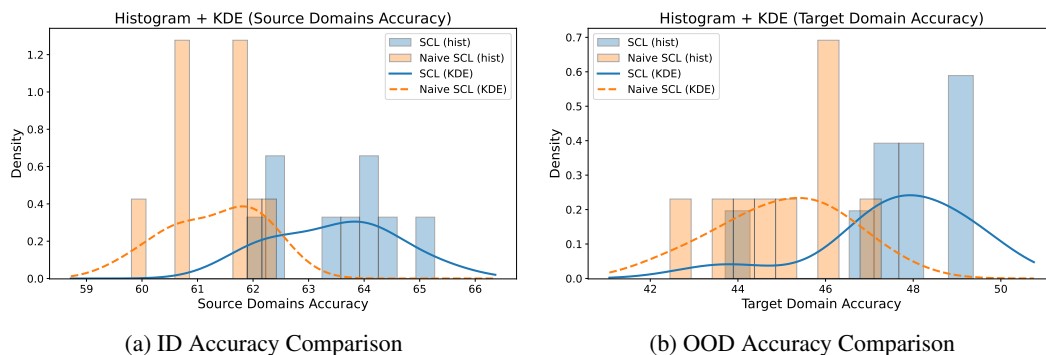

(a) ID Accuracy Comparison        (b) OOD Accuracy Comparison

Figure 2: Ablation Study: On dataset MNIST-C, based on ten runs with different random initializations, we plot the distribution of linear classification accuracy on source domains Figure 2a (ID accuracy) and target domains (OOD accuracy) Figure 2b based on SCL and Naive SCL representations. We present both frequency histograms and probability density functions smoothed via KDE. The experiment results demonstrate that SCL outperforms Naive SCL consistently.

(95% CI: [0.82, 3.50]). A paired $t$-test indicated statistical significance ($p = 0.0059$), which was further supported by the non-parametric Wilcoxon signed-rank test ($p = 0.0039$). Moreover, the linear classification accuracy on target domains, as shown in Figure 2b, demonstrates that SCL also outperforms Naive SCL stably. The mean performance of SCL is 47.62%, which exceeded that of Naive SCL (45.03%) by approximately 2.59% (95% CI: [0.97, 4.22]). A paired $t$-test indicated statistical significance ($p = 0.0063$), which was further supported by the non-parametric Wilcoxon signed-rank test ($p = 0.0195$). These results demonstrate that the significant improvement of SCL over Naive SCL is not due to chance.

## 6.6 Hyperparameters Analysis

In this section, we investigate the hyperparameters $(\lambda_1, \lambda_2)$ in Equation (2) and $(\tau_s, \tau_d)$ in Equation (3) Equation (4)'s effect on **SCL**'s learning performance. Specifically, we conduct experiments on MNIST-C, set domain 1 as the target domain, and domains 2, 3, and 4 as the source domains. We use the linear classification accuracy on both the source domains and the target domain as the evaluation metrics, which are defined as ID accuracy and OOD accuracy, respectively. In a three-dimensional space, we visualize the effects of different hyperparameter combinations $(\lambda_1, \lambda_2)$ and $(\tau_s, \tau_d)$ on ID accuracy and OOD accuracy using surfaces. Experiment results shown in Figure 3a and Figure 3b demonstrate that along the line where the $\lambda_2$ is 1.5, the surface exhibits a ridge-like shape. This fact indicates that $\lambda_2 = 1.5$ is a suitable hyperparameter. Excessively strong or excessively weak separation regularization hinders effective semantic extraction. In contrast, for $\lambda_1$, ID accuracy and OOD accuracy do not exhibit a clear variation trend. This suggests that the extraction of domain information is relatively insensitive to hyperparameter settings, indicating that domain-level representation learning is a comparatively straightforward task. Results, as demonstrated in Figure 3c and Figure 3d, indicate that the performance of **SCL** does not show a consistent trend with respect to changes in the hyperparameter values, indicating that **SCL** is robust to the temperature hyperparameter. In our setting, the default hyperparamters are $(\lambda_1, \lambda_2) = (1.0, 1.0)$ and $(\tau_s, \tau_d) = (0.1, 0.1)$.

## 6.7 Further Results

In this section, we investigate the performance of SCL under varying degrees of domain shift through experiments on RotatedMNIST. We set the unrotated images as the *target domain*, and use mixed datasets rotated by $\alpha$, $2\alpha$, and $3\alpha$ as the *source domains*, where $\alpha \in \{15°, 30°, 45°, 60°\}$. In this setting, larger rotation angles indicate more severe domain shifts. The experiment results in Figure 4a demonstrate that as the severity of domain shift increases, SCL consistently outperforms Naive SCL. Moreover, as the domain shift becomes larger, the target domain performance declines rapidly, whereas the source domain performance exhibits a more gradual degradation. In Figure 4b and Figure 4c, we present the learned representations $\mathbf{z}_s$ and $\mathbf{z_d}$ from the MNIST-C dataset through

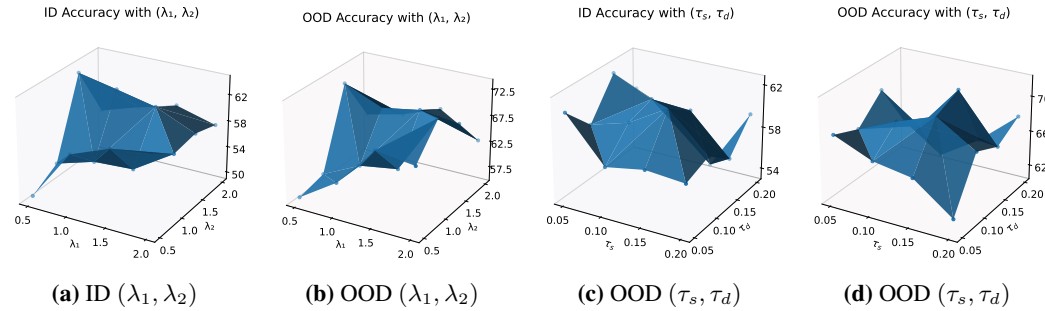

**(a)** ID $(\lambda_1, \lambda_2)$      **(b)** OOD $(\lambda_1, \lambda_2)$      **(c)** OOD $(\tau_s, \tau_d)$      **(d)** OOD $(\tau_s, \tau_d)$

**Figure 3:** Hyperparameters Analysis: On the MNIST-C dataset, we conduct experiments to analyze the hyperparameters' impact on SCL. Results in Figure 3a and Figure 3b demonstrate that SCL's performance is consistent with $\lambda_1$ when between 0.5 and 2.0 and is sensitive to $\lambda_2$. Overly large or overly small $\lambda_2$ degrades the performance of SCL. SCL achieves its best performance when $\lambda_2$ is about 1.5. Figure 3c and Figure 3d show the performance of SCL's with respect to the temperature parameters $\tau_s$ and $\tau_d$. SCL's performance does not exhibit a consistent variation trend with the temperature parameter.

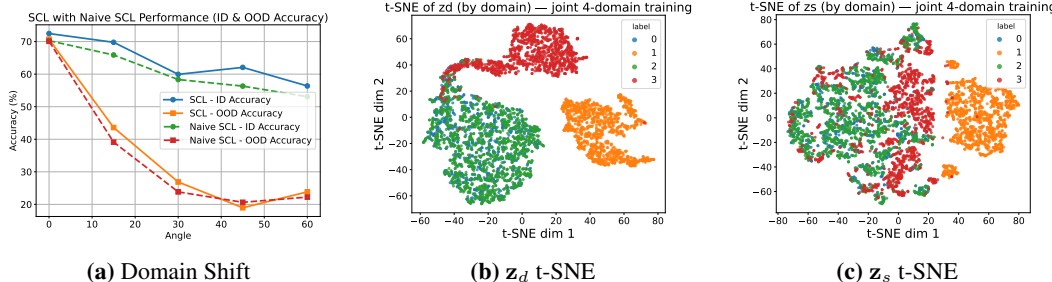

**(a)** Domain Shift      **(b)** $\mathbf{z}_d$ t-SNE      **(c)** $\mathbf{z}_s$ t-SNE

**Figure 4:** Further Analysis: The performance of SCL and Naive SCL is evaluated on RotatedMNIST under different rotation angles Figure 4a, illustrating their behavior under varying degrees of domain shift. In Figure 4b and Figure 4c, we present the learned representations $\mathbf{z}_s$ and $\mathbf{z_d}$ from the MNIST-C dataset through t-SNE visualization, where the points are colored according to domain labels. The results indicate that $\mathbf{z}_d$ forms distinct clusters according to domain labels. Further, domain 3 (Warped) is separated since the cut-and-distortion operation substantially modifies the shapes (semantics) of the digits.

t-SNE visualization, where the points are colored according to domain labels. This experiment is also based on MNIST-C, and all four domains are treated as training domains. The results show that $\mathbf{z}_d$ forms distinct clusters according to domain labels, while $\mathbf{z}_s$ appears to be entangled without clear separation. These indicate that the domain information is sufficiently extracted by $\mathbf{z}_d$, leaving little residual information in $\mathbf{z}_s$. As shown in the figure, the domain representations from domain 0 (Original) and domain 2 (Posterized) are intermixed in the two-dimensional t-SNE projection. This is because the two domains differ only slightly, sharing the same geometric structure and stroke patterns. Furthermore, the semantic representations of domain 3 (Warped) are clearly separated from those of the other domains. This separation can be attributed to the cut-and-distortion transformations present in domain 3 (Warped), which substantially modify the shapes of the digits. These shape variations are readily perceived as semantic differences.

# 7 ADDED EXPERIMENT RESULTS

## 7.1 RESULTS ON DOMAINNET

To validate SCL's performance on an extremely large dataset, we conduct experiments on Domain-Net, a multi-domain dataset with approximately 0.6 million images spanning 345 object categories collected from six visually diverse domains: Real, Clipart, Painting, Sketch, Infograph, and Quick-draw. Results are shown in Table 2.

**Table 2:** Performance comparison on DomainNet, with six domains as the target domain. Each cell reports ID Accuracy and OOD Accuracy (%).

| Methods | Clipart | | Infograph | | Painting | | Quickdraw | | Real | | Sketch | |
|---|---|---|---|---|---|---|---|---|---|---|---|---|
| | ID | OOD | ID | OOD | ID | OOD | ID | OOD | ID | OOD | ID | OOD |
| Naive SCL | 26.42 | 8.45 | 28.18 | 3.78 | 27.72 | 7.76 | 24.33 | 3.89 | 27.56 | 8.36 | 27.72 | 9.56 |
| SCL (ours) | 27.29 | 9.48 | 29.70 | 4.00 | 29.98 | 7.45 | 26.39 | 3.72 | 29.54 | 8.35 | 29.11 | 8.61 |

**Table 3:** Performance comparison on four MNIST-C domains. Each cell reports ID Accuracy and OOD Accuracy (%).

| Kernels | Original | | Solarized | | Posterized | | Warped | |
|---|---|---|---|---|---|---|---|---|
| | ID | OOD | ID | OOD | ID | OOD | ID | OOD |
| Guassian | 59.29 | 65.82 | 66.65 | 18.21 | 61.33 | 67.06 | 64.17 | 37.47 |
| Linear | 58.85 | 68.32 | 66.60 | 19.82 | 61.11 | 64.08 | 66.68 | 36.61 |
| Polynomial | 62.34 | 68.56 | 67.00 | 14.87 | 59.01 | 68.24 | 66.33 | 40.93 |
| Laplacian | 65.38 | 75.20 | 67.34 | 16.47 | 58.82 | 62.48 | 63.24 | 39.41 |
| Sigmoid | 59.83 | 73.52 | 68.78 | 11.11 | 60.74 | 66.56 | 68.38 | 40.85 |

## 7.2 ABLATION STUDY ON HSIC

In our previous default setting, we use the Gaussian kernel in HSIC's implementation. In this section, we conduct an ablation study on the impact of kernel type on SCL's learning performance. We conduct experiments on the dataset MNIST-c with different types of kernels, including the Gaussian kernel, the linear kernel, the Polynomial kernel, the Laplacian kernel, and the Sigmoid kernel. Learning performance is evaluated in both the training domain (ID) and target domains (OOD). Results in Table 3 demonstrate that across different kernel types, SCL maintains consistent performance, demonstrating its robustness to the disentangle technique.

## 8 CONCLUSIONS

We introduced Structured Contrastive Learning (SCL), a practical framework to learn disentangled semantic representation $\mathbf{z}_s$ and domain representation $\mathbf{z}_d$ simultaneously. In this framework, purified semantic information is captured by $\mathbf{z}_s$ without interference from domain shift, enabling robust self-supervised training with data from multiple domains and strong generalization ability to unseen domains. Theoretically, we have proved that SCL aligns with extracting semantic information and domain information at the same time, and shows that reducing $I(\mathbf{z}_s; \mathbf{z}_d)$ tightens bounds on target-domain risk, improving out-of-distribution generalization. Empirically, SCL delivers consistent gains over strong self-supervised and domain-generalization baselines on various datasets.

## 9 ETHICS STATEMENT

This work investigates structured contrastive learning under domain shift using only public or approval-based, de-identified datasets, in full compliance with their licenses and privacy requirements. No personally identifiable information was accessed or inferred, and the method does not enable tracking or identification of individuals. Any medical-related experiments are solely for research purposes; the models are not medical devices and should not inform clinical decisions without independent validation, regulatory review, and institutional ethics approval. Our study adheres to the ICLR Code of Ethics, and we report no conflicts of interest or external sponsorship that could bias the findings. To promote transparency and reproducibility, we will release the code and the ADNI dataset to the public upon acceptance.

## 10 REPRODUCIBILITY STATEMENT

Datasets descriptions, baseline methods, and evaluation metrics are clearly included in Section 6. To support transparency, we submitted our code as Supplementary Material. All of our baselines and SCL are run in the same hyperparameter setting. These materials ensure our reproducibility.

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

# A APPENDIX

## A.1 USE OF LLMS

In this paper, LLMs were used solely for writing polishing; all substantive writing, ideas, and content remain human-authored.

## A.2 PROOF FOR THEOREMS

**Theorem 4.** *Let* $\mathbf{S}$ *be a latent semantic factor. Assume the two-view conditional independence* $\mathbf{z}_s \perp \mathbf{z}_s^+ \mid \mathbf{S}$. *Moreover, we assume that* $\mathbf{z}_s$ *and* $\mathbf{z}_s^+$ *are sufficient statistics for* $\mathbf{S}$, *then we have:*

$$\mathbf{argmax}_\theta I(\mathbf{z}_s; \mathbf{z}_s^+) = \mathbf{argmax}_\theta I(\mathbf{z}_s; \mathbf{S}), \tag{10}$$

*where* $\theta$ *are parameters for encoder* $f_s(\cdot)$, $f_d(\cdot)$ *and* $\phi(\cdot)$.

*Proof.* By the chain rule, $I(\mathbf{z}_s; \mathbf{z}_s^+, \mathbf{S}) = I(\mathbf{z}_s; \mathbf{S}) + I(\mathbf{z}_s; \mathbf{z}_s^+ \mid \mathbf{S}) = I(\mathbf{z}_s; \mathbf{z}_s^+) + I(\mathbf{z}_s; \mathbf{S} \mid \mathbf{z}_s^+)$. The assumption $\mathbf{z}_s \perp \mathbf{z}_s^+ \mid \mathbf{S}$ yields $I(\mathbf{z}_s; \mathbf{z}_s^+ \mid S) = 0$, giving the stated identity and the inequality by nonnegativity of conditional mutual information. If both $\mathbf{z}_s$ and $\mathbf{z}_s^+$ are sufficient for $S$ (so $I(\mathbf{z}_s; \mathbf{S}) = I(\mathbf{z}_s^+; \mathbf{S}) = H(\mathbf{S})$ and $I(\mathbf{z}_s; \mathbf{S} \mid \mathbf{z}_s^+) = 0$), the identity forces $I(\mathbf{z}_s; \mathbf{z}_s^+) = H(\mathbf{S})$ at the same parameters; thus the two maximization problems have the same global maximum and coincide in their global maximizers (modulo measure-zero reparameterizations). $\square$

**Theorem 5.** *Let* $d$ *be a domain label. Assume the two-view conditional independence* $\mathbf{z}_d \perp \mathbf{z}_d^+ \mid d$. *Moreover, we assume that* $\mathbf{z}_d$ *and* $\mathbf{z}_d^+$ *are sufficient statistics for* $d$, *then we have:*

$$\mathbf{argmax}_\theta I(\mathbf{z}_d; \mathbf{z}_d^+) = \mathbf{argmax}_\theta I(\mathbf{z}_d; \mathbf{D}), \tag{11}$$

*where* $\theta$ *are parameters for encoder* $f_s(\cdot)$, $f_d(\cdot)$ *and* $\phi(\cdot)$.

*Proof.* Apply the chain rule to $I(\mathbf{z}_d; \mathbf{z}_d^+, d)$: $I(\mathbf{z}_d; \mathbf{z}_d^+, d) = I(\mathbf{z}_d; d) + I(\mathbf{z}_d; \mathbf{z}_d^+ \mid d) = I(\mathbf{z}_d; \mathbf{z}_d^+) + I(\mathbf{z}_d; d \mid \mathbf{z}_d^+)$. Under $\mathbf{z}_d \perp \mathbf{z}_d^+ \mid d$, we have $I(\mathbf{z}_d; \mathbf{z}_d^+ \mid d) = 0$, which yields the stated identity and inequality. If both $\mathbf{z}_d$ and $\mathbf{z}_d^+$ are sufficient for $d$ (so $I(\mathbf{z}_d; d) = I(\mathbf{z}_d^+; d) = H(\mathbf{D})$ and $I(\mathbf{z}_d; d \mid \mathbf{z}_d^+) = 0$), the identity forces $I(\mathbf{z}_d; \mathbf{z}_d^+) = H(d)$ at the same parameters; the optimization equivalence follows as in Theorem 1. $\square$

Let $d \in \{1, 2\}$ indicate whether a sample comes from the source domain $\mathcal{D}_1$ or the target domain $\mathcal{D}_2$. Write $\mathbf{z}_s = f_s(\phi(\mathbf{x})) \in \mathbb{R}^p$ and $\mathbf{z}_d = f_d(\phi(\mathbf{x}))$, and consider a linear probe $h_w(\mathbf{z}_s) = \mathbf{w}^\top \mathbf{z}_s$ with $\|\mathbf{w}\| \leq W$. Denote the class label by $y$, the population risk on domain $k \in \{1, 2\}$ by

$$R_k(w) = \mathbb{E}_{(\mathbf{z}_s, y) \sim \mathcal{D}_k}\big[\ell\big(y, \mathbf{w}^\top \mathbf{z}_s\big)\big],$$

and the empirical risk on $\mathcal{D}_1$ by $\hat{R}_1(\mathbf{w})$ with $n$ samples. We assume $\|\mathbf{z}_s\| \leq B$ almost surely on both domains and that $|\partial_u \ell(y, u)| \leq 1$, where $u = \mathbf{w}^\top \mathbf{z}_s$.

**Assumption 4** (Prior alignment or reweighting). *Either* $p_1(y) = p_2(y)$ *holds, or we evaluate* $R_1$ *and* $\hat{R}_1$ *under importance weighting so that the effective class prior matches* $p_2(y)$.

**Assumption 5** (Kernel and metric control). *Let* $\mathsf{k} : \mathbb{R}^p \times \mathbb{R}^p \to \mathbb{R}$ *be a bounded, sufficiently smooth kernel (e.g., a Matérn kernel inducing a Sobolev RKHS of order* $s > p/2 + 1$) *on a compact support of* $\mathbf{z}_s$. *There exists a constant* $C_\mathsf{k} > 0$ *such that for any probability measures* $P, Q$ *on the support,*

$$W_1(P, Q) \leq C_\mathsf{k} \, \mathrm{MMD}_\mathsf{k}(P, Q).$$

For each class $y_i$, let $P_k^{y_i}$ be the conditional distribution of $\mathbf{z}_s$ given $(y = y_i, d = k)$. Define the class-conditional kernel discrepancy

$$\Delta_{\mathrm{MMD}} \triangleq \sum_{y_i} p_2(y_i) \, \mathrm{MMD}_\mathsf{k}\big(P_2^{y_i}, P_1^{y_i}\big).$$

**Theorem 6** (Transfer bound via class-conditional MMD). *Under the above assumptions, for any* $\|\mathbf{w}\| \leq W$,

$$R_2(\mathbf{w}) \leq R_1(\mathbf{w}) + W \, C_\mathsf{k} \, \Delta_{\mathrm{MMD}}.$$

*Proof.* By class-wise decomposition and the Kantorovich–Rubinstein duality,

$$R_2(\mathbf{w}) - R_1(\mathbf{w}) = \sum_{y_i} p_2(y_i)\Big(\mathbb{E}_{\mathbf{z} \sim P_2^{y_i}}\ell(y, \mathbf{w}^\top \mathbf{z}) - \mathbb{E}_{\mathbf{z} \sim P_1^{y_i}}\ell(y, \mathbf{w}^\top \mathbf{z})\Big).$$

For fixed $y_i$, the function $g_{y_i}(\mathbf{z}) = \ell(y, \mathbf{w}^\top \mathbf{z})$ is $\|\mathbf{w}\|$-Lipschitz in $\mathbf{z}$ because $|\partial_u \ell| \leq 1$ and $\nabla_{\mathbf{z}} g_{y_i} = \ell'(\mathbf{w}^\top \mathbf{z})$. Hence

$$\left| \mathbb{E}_{P_2^{y_i}} g_{y_i} - \mathbb{E}_{P_1^{y_i}} g_{y_i} \right| \; \leq \; \|\mathbf{w}\|\, W_1(P_2^{y_i}, P_1^{y_i}) \; \leq \; \|\mathbf{w}\|\, C_{\mathsf{k}}\; \mathrm{MMD}_{\mathsf{k}}(P_2^{y_i}, P_1^{y_i}).$$

Summing over $y_i$ and using $\|\mathbf{w}\| \leq W$ gives the claim. $\qquad\square$

**Theorem 7** (From population to empirical source risk). *With probability at least $1 - \delta$ over the draw of $n$ source samples from $\mathcal{D}_1$, the following holds simultaneously for all $\|\mathbf{w}\| \leq W$:*

$$R_1(\mathbf{w}) \; \leq \; \hat{R}_1(\mathbf{w}) \; + \; 2\,\frac{WB}{\sqrt{n}} \; + \; 3\sqrt{\frac{\ln(2/\delta)}{2n}}.$$

*Proof.* The class $\{\mathbf{z} \mapsto \mathbf{w}^\top \mathbf{z} : \|\mathbf{w}\| \leq W\}$ has Rademacher complexity at most $WB/\sqrt{n}$ under $\|\mathbf{z}\| \leq B$. Because $\ell(y, \cdot)$ is 1-Lipschitz in the margin, the contraction inequality implies the stated bound. $\qquad\square$

**Theorem 8** (From $I(\mathbf{z}_s; \mathbf{z}_d)$ to $\Delta_{\mathrm{MMD}}$). *Assume that, given $Y$, the domain signal is sufficiently captured by $\mathbf{z}_d$ so that the Markov chain $d \to \mathbf{z}_d \to \mathbf{z}_s$ holds conditionally on $y$. Then there exists a constant $C > 0$ (depending on $\mathsf{k}$ and the support) such that*

$$\Delta_{\mathrm{MMD}} \; \leq \; C\,\sqrt{I(\mathbf{z}_s; d \mid y)} \; \leq \; C\,\sqrt{I(\mathbf{z}_s; \mathbf{z}_d \mid y)} \; \leq \; C\,\sqrt{I(\mathbf{z}_s; \mathbf{z}_d)}.$$

*Proof.* For the binary domain variable $D \in \{1, 2\}$, the conditional mutual information satisfies $I(\mathbf{z}_s; d \mid y = y_i) = \mathrm{JS}\big(P_2^{y_i} \| P_1^{y_i}\big)$ (conditional Jensen–Shannon divergence). Standard Pinsker-type inequalities yield $\mathrm{TV}(P_2^{y_i}, P_1^{y_i}) \leq c_1\sqrt{I(\mathbf{z}_s; d \mid y = y_i)}$ for a universal constant $c_1$, and for bounded, characteristic kernels there exists $c_2$ with $\mathrm{MMD}_{\mathsf{k}}(P_2^{y_i}, P_1^{y_i}) \leq c_2\,\mathrm{TV}(P_2^{y_i}, P_1^{y_i]})$. Combining and averaging over $y_i$ gives $\Delta_{\mathrm{MMD}} \leq C\sqrt{I(\mathbf{z}_s; D \mid y)}$ for $C = c_1 c_2$. By the data processing inequality under $d \to \mathbf{z}_d \to \mathbf{z}_s$ given $y$, $I(\mathbf{z}_s; d \mid y) \leq I(\mathbf{z}_s; \mathbf{z}_d \mid y) \leq I(\mathbf{z}_s; \mathbf{z}_d)$, which completes the proof. $\qquad\square$

**Corollary 1** (Main bound; minimizing $I(\mathbf{z}_s; \mathbf{z}_d)$ improves target generalization). *Let $\hat{\mathbf{w}} = \mathrm{argmin}_{\|\mathbf{w}\| \leq W} \hat{R}_1(\mathbf{w})$ be the linear probe trained on $\mathcal{D}_1$. Under Assumptions 1–2 and the condition of Theorem 8, with probability at least $1 - \delta$,*

$$R_2(\hat{\mathbf{w}}) \; \leq \; \hat{R}_1(\hat{\mathbf{w}}) \; + \; 2\,\frac{WB}{\sqrt{n}} \; + \; 3\sqrt{\frac{\ln(2/\delta)}{2n}} \; + \; W\,C_{\mathsf{k}}\,C\,\sqrt{I(\mathbf{z}_s; \mathbf{z}_d)}.$$

*Consequently, any training strategy that decreases $I(\mathbf{z}_s; \mathbf{z}_d)$ on $\mathcal{D}_1$ (while keeping $W, B$ controlled) provably tightens the upper bound on the target risk $R_2(\hat{\mathbf{w}})$ on $\mathcal{D}_2$.*

*Proof.* Combine Theorem 6 with Theorem 7, then substitute Theorem 8. $\qquad\square$

**Remark.** If the class priors are not aligned and reweighting is not used, an extra additive term depending on $|p_2(y) - p_1(y)|$ appears in Theorem 6; this does not affect the dependence on $\Delta_{\mathrm{MMD}}$ or $I(\mathbf{z}_s; \mathbf{z}_d)$ and is omitted here for clarity.

## A.3 EXPERIEMNTAL DETAILS

### A.3.1 DATASETS

To validate our proposed method, we conducted experiments on three public image datasets and one private medical dataset. The details of these five datasets and their preprocessing procedures are as follows.

- **MNIST-C** We construct a four-domain, synthetic dataset based on MNIST, a classic dataset of handwritten digits with semantic labels ranging from 0 to 9. The original corpus, comprising 60,000 training and 10,000 test images (28×28 grayscale digits across ten classes), is evenly partitioned into four domains, resulting in 12,000 training, 1,500 validation, and 1,500 test images per domain. Each domain carries a simple, human-interpretable "style": Domain 1 (Original) preserves the original appearance; Domain 2 (Solarized) applies exposure solarization with pronounced sharpening; Domain 3 (Posterized) reduces tonal detail (posterization) and boosts overall contrast; Domain 4 (Warped) introduces perspective warping together with a modest shear, creating geometric distortion while keeping the digit identity unchanged. We follow the classical setting and adopt 29×29 grayscale images.

- **Rotated MNIST** Another four-domain dataset based on MNIST. Each domain contains 12,000 training, 1,500 validation, and 1,500 test images. In this setting, domains are constructed by rotation angles. Domains 1, 2, 3, and 4 correspond to images rotated clockwise by 0°, 30°, 60°, and 90°, respectively. We follow the classical setting and adopt 29×29 grayscale images.

- **PACS** The PACS dataset is a domain generalization benchmark with four visually distinct domains (Photo, Art Painting, Cartoon, Sketch) and seven object categories (dog, elephant, giraffe, guitar, horse, house, person), totaling $\sim 10,000$ images. By mixing natural photographs with artist-rendered depictions and line drawings, PACS induces substantial shifts in texture, contour, and abstraction while keeping labels fixed, making it a strong testbed for learning under domain shift. Images have varied resolutions and aspect ratios; in practice, they are resized and normalized for the chosen backbone (e.g., $96 \times 96$ in our setting).

- **ADNI** This dataset is obtained from the Alzheimer's Disease Neuroimaging Initiative (ADNI) database (adni.loni.usc.edu). ADNI was launched in 2003 as a public-private partnership. One of the primary goals of the Alzheimer's Disease Neuroimaging Initiative (ADNI) has been to evaluate whether multimodal neuroimaging techniques, such as magnetic resonance imaging (MRI) and positron emission tomography (PET), can be integrated to track the progression of mild cognitive impairment (MCI) and early Alzheimer's disease (AD). ADNI has recruited cognitively normal individuals (CN), those with MCI, and those with dementia or AD. The ADNI data were collected from 39 sites, representing 39 domains in our study. T1-weighted (T1w) MRI data from 3T MRI scanners were included in this study. All T1w images underwent automated quality control through MRIQC. For all selected T1w images that passed the quality check, cross-sectional image processing was performed using FreeSurfer Version 7.1.1 (https://surfer.nmr.mgh.harvard.edu/). Region of interest (ROI)-specific cortical thickness (CT) values were extracted from the automated anatomical parcellation using the Desikan-Killiany Atlas (Desikan et al., 2006).

### A.3.2 BASELINE METHODS

To validate SCL's performance, we compare it with a diverse set of current methods towards Self-Supervised Learning and Domain Generalization.

- *Self-Supervised Learning Methods*

    **SimCLR** Chen et al. (2020a) is one of the most widely used contrastive learning methods, and is well-known for its robustness in extracting semantics from redundant and noisy training samples without supervision.

    **Moco** He et al. (2020) is a well-known contrastive learning method that uses a momentum-updated encoder and a queue of negative samples. By maintaining consistent representations and an extensive dynamic dictionary, MoCo effectively learns representations without labels.

    **Naive SCL** is a framework with the same structure as our proposed **SCL**. The only difference between **Naive SCL** and **SCL** is that we froze the optimization of the domain head and the disentanglement term when training **Naive SCL**, only optimizing the semantic head Chen et al. (2020a). By comparing with **Naive SCL**, we can exclude the influence of model architecture on the performance of **SCL** and verify whether our proposed training paradigm truly improves performance.

- *Domain Generalization Methods*

**Sagnet** Nam et al. (2021) is a domain generalization method that suppresses style cues—via feature-level style randomization and adversarial training—so models rely on content and generalize robustly to unseen domains.

**SSRL-MD** Feng et al. (2019) trains a single self-supervised encoder on multi-domain data, combining a gradient-reversal domain classifier to remove domain cues with a contrastive JSD-based term to preserve within-domain information. Applied atop standard pretext tasks (e.g., RotNet/AET), it improves cross-domain generalization.

**DDM** Kalibhat et al. (2023) adds a small domain-coded prefix and adversarially enforces domain-invariance on the rest of the features, with optional robust clustering for unlabeled domains—yielding stronger cross-domain SSL representations.

### A.4 FURTHER RELATED WORKS: SELF-SUPERVISED REPRESENTATION LEARNING

Self-supervised learning seeks to learn discriminative and generalizable features from rich unlabeled data. The effectiveness of self-supervised learning largely depends on the design of the pretext task, which is optimized with a dedicated loss function. Broadly, pretext tasks can be grouped into three categories: context-based methods, generative algorithms, and contrastive learning. Context-based methods include tasks such as rotation Gidaris et al. (2018), colorization Larsson et al. (2016); Zhang et al. (2017b), and jigsaw solving Goyal et al. (2019). Generative algorithms mainly focus on masked object modeling, which learn representations by reconstructing missing or corrupted parts of the input. Representative examples include masked image modeling approaches such as MAE He et al. (2022), as well as masked language modeling methods like BERT Devlin et al. (2019) and GPT Brown et al. (2020). In contrastive learning, a prominent line of work focuses on instance discrimination, where the goal is to pull positive samples closer together while pushing negative samples apart in the latent space. Notable contributions in this direction include MoCo He et al. (2020), which leverages the InfoNCE loss function as a form of contrastive loss. Subsequent influential works, such as MoCo v2 Chen et al. (2020b) and SimCLR Chen et al. (2020a), further refined this line of research. Our proposed self-supervised learning framework is built on the instance discrimination-based contrastive learning approach.

