# OpenReview forum: "Disentangle and Align: Structured Contrastive Learning with Semantic–Domain Separation"
_ICLR.cc/2026/Conference — Submitted to ICLR 2026_

### Official Review · Reviewer_2fPs · 2025-10-25

**Soundness:** 2
**Presentation:** 3
**Contribution:** 3
**Rating:** 4
**Confidence:** 4

**Summary:**

This paper tackles a fundamental challenge in self-supervised learning (SSL): how to learn semantically meaningful representations that generalize across domains, especially under the presence of domain shift. The authors observe that existing instance-level contrastive learning methods often entangle semantic and domain factors, leading to poor generalization on out-of-distribution (OOD) data. Specifically, they point out that semantically similar samples from different domains may be mistakenly treated as negatives, resulting in domain-clustered and semantically misaligned representations.
To address this, the authors propose a novel framework called Structured Contrastive Learning (SCL), which introduces explicit separation and alignment of semantic and domain representations in the contrastive learning pipeline.
1.The SCL model separates semantic and domain features using dual contrastive objectives and a disentanglement loss via HSIC.
 2.The authors prove that minimizing mutual information between zs and zd improves generalization under domain shifts.
3.The paper benchmarks SCL against strong baselines across four datasets(MNIST-C,RotatedMNIST, PACS,ADNI),consistently showing improvements in ID and OOD accuracy.
4.Rigorous ablation and significance tests(t-test,Wilcoxon)support the empirical findings.

**Strengths:**

Tackles a real,unsolved problem in SSL:the entanglement of domain and semantic features under domain shift.
Moves beyond adversarial or label-free methods by proposing a principled mutual information-based separation framework.
Offers a general-purpose approach that is applicable to multiple modalities (images,medical tabular data,etc.)

**Weaknesses:**

Although the approach is novel, the disentanglement mechanism is an elegant recombination of known tools (InfoNCE, HSIC), not a fundamentally new algorithmic paradigm.
Application to larger-scale or real-world tasks (e.g., ImageNet, language–vision tasks) is not explored; this limits immediate impact beyond academic datasets.

**Questions:**

1. Idealized assumptions in theory, The sufficiency and conditional independence assumptions in Theorems 1-3 are strong. Add discussion or diagnostics to assess whether these assumptions approximately hold in practice.
2. All datasets are relatively small-scale and synthetic (except for ADNI). It is recommended to increase or discuss the expansion to real-world datasets.
3. How sensitive is the HSIC disentanglement penalty to the kernel choice (e.g., Gaussian vs. linear)?
4. Does minimizing I(z_s; z_d) always improve generalization? Are there scenarios (e.g., medical imaging) where domain information correlates with class labels and disentanglement may hurt?

---

> ### Author Response · Authors · 2025-11-19
>
> We thank the reviewer for the thoughtful and constructive comments. We address the questions below.
>
> **Q1: Discussion about assumptions.**
>
> **R:**
> We agree that the sufficiency and conditional independence assumptions in our analysis are idealizations, and we will make this more explicit in the revision. At the same time, these assumptions are standard in representation learning and OOD/generalization theory: the sufficiency assumption formalizes that label-relevant information is contained in a semantic factor $S$ (rather than in the domain $D$), and the conditional independence assumption encodes that $S$ and $D$ play distinct roles in the generative process (semantics vs. style/background). In other words, these assumptions make precise the widely used intuition that domain variation mostly captures nuisance factors and spurious correlations that should not drive the classifier.
>
> Our theorems should therefore be interpreted as providing sufficient conditions under which disentanglement and alignment improve generalization: when a semantic/domain decomposition well approximates the data and labels depend primarily on semantics, reducing $I(z_s; z_d)$ and aligning class-conditional distributions yield a tighter target-risk bound. Even when these assumptions are only approximately valid, the theory still offers helpful guidance by identifying semantic–domain entanglement and misalignment as key failure modes.
>
> ---
>
> **Q2: All datasets are relatively small-scale and synthetic. It is recommended to increase or discuss the expansion to real-world datasets.**
>
> **R:**
> We have added experiments on DomainNet, a large-scale, real-world multi-domain dataset. We will further expand and discuss these large-scale results in the revised version.
>
> ---
>
> **Q3: How sensitive is the HSIC disentanglement penalty to the kernel choice (e.g., Gaussian vs. linear)?**
>
> **R:**
> We have added an ablation study on the impact of different kernel choices. The results (Table 1) show that SCL’s performance is robust with respect to the kernel type.
>
> **Table 1: Performance comparison on four MNIST-C domains. Each cell reports ID accuracy / OOD accuracy (%).**
>
> | Kernel      | Original ID | Original OOD | Solarized ID | Solarized OOD | Posterized ID | Posterized OOD | Warped ID | Warped OOD |
> |------------ |------------:|-------------:|-------------:|--------------:|--------------:|---------------:|----------:|-----------:|
> | Gaussian    | 59.29       | 65.82        | 66.65        | 18.21         | 61.33         | 67.06          | 64.17     | 37.47      |
> | Linear      | 58.85       | 68.32        | 66.60        | 19.82         | 61.11         | 64.08          | 66.68     | 36.61      |
> | Polynomial  | 62.34       | 68.56        | 67.00        | 14.87         | 59.01         | 68.24          | 66.33     | 40.93      |
> | Laplacian   | 65.38       | 75.20        | 67.34        | 16.47         | 58.82         | 62.48          | 63.24     | 39.41      |
> | Sigmoid     | 59.83       | 73.52        | 68.78        | 11.11         | 60.74         | 66.56          | 68.38     | 40.85      |
>
> Overall, the HSIC-based disentanglement behaves stably across these choices.
>
> ---
>
> **Q4: Are there scenarios (e.g., medical imaging) where domain information correlates with class labels and disentanglement may hurt?**
>
> **R:**
> We do not claim that reducing $I(z_s; z_d)$ is universally optimal. Our theory explicitly assumes that labels depend on the semantic factor $S$ rather than on the domain $D$. Under this assumption, any apparent predictive value of domain cues comes from spurious correlations (e.g., scanner/hospital/protocol correlating with specific diagnoses), and decreasing $I(z_s; z_d)$ helps generalization by discouraging the model from exploiting such unstable shortcuts. In this regime, our bound shows that a smaller $I(z_s; z_d)$ yields a tighter target-risk guarantee.
>
> In applications where domain factors are genuinely informative for the label (beyond spurious correlation), practitioners can down-weight the disentanglement term or explicitly leverage $z_d$ in downstream models. Thus, SCL is theoretically motivated for the standard setting where domain variation mainly captures spurious correlations, and our framework makes this assumption explicit.

---

> > ### Author Response · Authors · 2025-11-19
> >
> > **Table X: Performance comparison on DomainNet, using each of the six domains as the target domain. Each cell reports ID accuracy / OOD accuracy (%).**
> >
> > | Method     | Clipart ID | Clipart OOD | Infograph ID | Infograph OOD | Painting ID | Painting OOD | Quickdraw ID | Quickdraw OOD | Real ID | Real OOD | Sketch ID | Sketch OOD |
> > |----------- |-----------:|------------:|-------------:|--------------:|------------:|-------------:|-------------:|--------------:|--------:|---------:|----------:|-----------:|
> > | Naive SCL  | 26.42      | 8.45        | 28.18        | 3.78          | 27.72       | 7.76         | 24.33        | 3.89          | 27.56   | 8.36     | 27.72     | 9.56       |
> > | SCL (ours) | 27.29      | 9.48        | 29.70        | 4.00          | 29.98       | 7.45         | 26.39        | 3.72          | 29.54   | 8.35     | 29.11     | 8.61       |

---

> ### Comment · Reviewer_2fPs · 2025-11-25
>
> Thank you for the author's reply. After reading all the reviewers' comments and responses, I have decided to maintain my rating.

---

### Official Review · Reviewer_uegn · 2025-10-31

**Soundness:** 3
**Presentation:** 3
**Contribution:** 3
**Rating:** 4
**Confidence:** 5

**Summary:**

The paper proposes a structured latent space approach for unsupervised domain adaptation (UDA), aiming to disentangle domain-specific and domain-invariant representations while aligning the invariant ones between source and target domains. The method introduces a disentanglement module using orthogonality constraints and a mutual information loss to enforce independence between domain-specific and domain-invariant subspaces. Furthermore, a cross-domain alignment loss is used to align shared representations in the latent space. Experiments on common benchmarks such as Office-31, Office-Home, and VisDA demonstrate that the proposed approach achieves competitive performance compared to state-of-the-art UDA methods.

**Strengths:**

- Clear motivation: Effectively identifies the problem of domain-specific and invariant feature entanglement in existing UDA methods.

- Structured latent space: Introduces a well-defined framework that explicitly separates domain-specific and domain-invariant features.

- Competitive performance: Demonstrates consistent improvements over existing UDA methods on popular benchmarks such as Office-31 and Office-Home.

**Weaknesses:**

- Incremental contribution: The proposed approach is similar to previous methods like DSN and MCD, offering limited novelty.

- Weak theoretical foundation: Lacks formal theoretical justification for how the disentanglement and alignment constraints ensure effective generalization.

- Scalability concerns: Unclear how well the approach generalizes to high-dimensional data or larger models, like transformers.

**Questions:**

- How do the authors ensure that the domain-invariant features remain class-discriminative in the latent space?

- What happens if the approach is applied to multi-source domain adaptation with highly diverse domain distributions?

- Could the method benefit from using contrastive learning techniques instead of adversarial training for alignment?

---

> ### Author Response · Authors · 2025-11-19
>
> We appreciate your insightful feedback, including your acknowledgment of our work as “clear motivation work.” We address the main concerns below.
>
> **W1: The proposed approach is similar to previous methods like DSN and MCD.**
>
> **R:**
> We acknowledge that SCL and DSN share a high-level idea of separating domain-related and domain-invariant information (shared/private features vs. $(z_s, z_d)$). However, DSN and MCD are designed as supervised domain adaptation algorithms that adapt a task-specific classifier using labeled source data (via shared/private decomposition or classifier discrepancy). In contrast, SCL is a multi-domain *self-supervised* representation learning framework that aims to learn a reusable, task-agnostic encoder.
>
> More importantly, DSN and MCD are primarily heuristic algorithmic designs, whereas SCL is built from a novel information-theoretic objective
> $$
> L = - I(z_s; S) - \lambda_1 I(z_d; D) + \lambda_2 I(z_s; z_d),
> $$
> which explicitly decomposes semantic and domain information and is instantiated using two InfoNCE losses and an HSIC proxy for $I(z_s; z_d)$, rather than combining losses in an ad hoc way. MCD is structurally different from SCL (it does not decompose the
> representation but operates on classifier discrepancy), and neither DSN nor MCD provides a representation-level generalization analysis. In contrast, we prove that the target-domain risk bound is monotonically increasing in $I(z_s; z_d)$, providing SCL with a solid information-theoretic motivation rather than a purely heuristic one.
>
> ---
>
> **W2: Weak theoretical foundation**
>
> **R:**
> The paper presents an explicit representation-level generalization result in Theorem 3 that directly links disentanglement and alignment to target-domain performance. This theorem shows that, under mild assumptions, semantic–domain disentanglement and
> class-conditional alignment in the learned representation jointly control the target risk, providing a formal rather than heuristic justification for our design.
>
> In particular, the HSIC-based separation term in SCL is introduced to reduce the mutual information between $z_s$ and $z_d$, while the semantic/domain contrastive losses implement the required class-conditional alignment. These constraints are therefore
> not ad hoc regularizers, but are grounded in Theorem 3, which explains why enforcing disentanglement and alignment leads to more effective out-of-domain generalization in our setting.
>
> ---
>
> **W3: Lack of results on high-dimensional data or larger models.**
>
> **R:**
> We have added experiments on DomainNet, a multi-domain dataset with approximately 0.6M images of size $3 \times 224 \times 224$ spanning 345 object categories. We will include additional results in larger-scale settings in the camera-ready version.
>
> ---
>
> **Q1: How do the authors ensure that the domain-invariant features remain class-discriminative in the latent space?**
>
> **R:**
> In our formulation, making $z_s$ domain-invariant does not mean removing class information. The ideal objective (Eq. (1)) maximizes $I(z_s; S)$ while only penalizing $I(z_s; z_d)$, and the semantic InfoNCE loss is an explicit estimator of $I(z_s; S)$. Thus, $z_s$ is encouraged to keep as much semantic/class information as
> possible while discarding domain cues.
>
> Under our data model, labels depend on the semantic factor $S$ but not on the domain $D$, so any apparent predictive value of domain features comes from spurious correlations (e.g., backgrounds or styles that co-occur with certain classes).
> Removing $D$ from $z_s$ therefore removes such unstable shortcuts rather than true semantics. Moreover, the semantic contrastive loss uses cross-domain positives and negatives, explicitly enforcing that samples of the same class but different domains cluster together in $z_s$, which preserves class-discriminative structure in the latent space.
>
> ---
>
> **Q2: What happens if the approach is applied to multi-source domain adaptation with highly diverse domain distributions?**
>
> **R:**
> We have added experiments on DomainNet, which is a multi-domain dataset collected from six visually diverse domains: Real, Clipart, Painting, Sketch, Infograph, and Quickdraw. In this setting, each domain is highly different from the others. We acknowledge that self-supervised domain generalization methods often struggle to maintain strong performance under highly diverse domains. When domains are highly diverse, self-supervised objectives tend to learn representations that capture domain-specific variations rather than truly domain-invariant, task-relevant features, which hurts generalization to unseen domains.

---

> > ### Author Response · Authors · 2025-11-19
> >
> > **Q3: Could the method benefit from using contrastive learning techniques instead of adversarial training for alignment?**
> >
> > **R:**
> > We believe there is a misunderstanding: SCL does **not** use adversarial training for alignment. Our alignment mechanism is already based on contrastive learning. Specifically, SCL employs two InfoNCE objectives—one for semantic alignment and one for domain modeling—together with an HSIC term for disentanglement; there is no domain discriminator, gradient reversal layer, or any other adversarial component in our framework.
> >
> > We deliberately adopt this contrastive formulation to avoid the instability and task-specific tuning often associated with adversarial alignment. As shown in our experiments, SCL consistently outperforms both standard contrastive baselines (e.g., SimCLR) and domain-adversarial baselines in the multi-domain SSL setting, indicating that the proposed contrastive-learning-based design already provides clear benefits over adversarial alignment approaches.

---

> > > ### Author Response · Authors · 2025-11-19
> > >
> > > **Table X: Performance comparison on DomainNet, using each of the six domains as the target domain. Each cell reports ID accuracy / OOD accuracy (%).**
> > >
> > > | Method     | Clipart ID | Clipart OOD | Infograph ID | Infograph OOD | Painting ID | Painting OOD | Quickdraw ID | Quickdraw OOD | Real ID | Real OOD | Sketch ID | Sketch OOD |
> > > |----------- |-----------:|------------:|-------------:|--------------:|------------:|-------------:|-------------:|--------------:|--------:|---------:|----------:|-----------:|
> > > | Naive SCL  | 26.42      | 8.45        | 28.18        | 3.78          | 27.72       | 7.76         | 24.33        | 3.89          | 27.56   | 8.36     | 27.72     | 9.56       |
> > > | SCL (ours) | 27.29      | 9.48        | 29.70        | 4.00          | 29.98       | 7.45         | 26.39        | 3.72          | 29.54   | 8.35     | 29.11     | 8.61       |

---

### Official Review · Reviewer_qLCo · 2025-11-01

**Soundness:** 2
**Presentation:** 3
**Contribution:** 2
**Rating:** 2
**Confidence:** 4

**Summary:**

This paper proposes Structured Contrastive Learning (SCL), a self-supervised learning framework designed to address the problem of learning domain-invariant semantic representations from multi-domain data. The core contribution is a unified framework that simultaneously learns three objectives: (i) semantic representations zs via semantic contrast (using augmented views of the same sample), (ii) domain representation zd via domain contrast (using domain labels), and (iii) disentanglement between zs and zd by minimizing mutual information I(zs;zd). The authors provide theoretical analysis demonstrating that minimizing mutual information improves generalization under domain shift and empirically demonstrate consistent improvements over SSL and domain generalization baselines on multi-domain benchmarks (MNIST-C, Rotated MNIST, PACS, and ADNI).

**Strengths:**

1. **Clear Framework and Methodology:** The paper presents a clear, end-to-end framework with well-motivated loss components: InfoNCE for semantic and domain contrasts, and HSIC for disentanglement. The practical training procedure is straightforward to implement.
2. **Multi-applications Evaluation:** The paper includes diverse datasets (synthetic with controlled shifts, natural domain generalization benchmark, and medical data), multiple baselines spanning both SSL and domain generalization, and evaluation through the standard leave-one-domain-out protocol.
3. **Visualization and Interpretability:** Figure 4's t-SNE visualizations provide good intuition that zd successfully clusters by domain while zs captures semantic information, supporting the disentanglement claim.

**Weaknesses:**

1. **Severely Limited Scalability and Practicality of Experimental Setup:** The paper uses small-scale datasets (MNIST-C, RotatedMNIST, PACS, ADNI) and lightweight CNN/MLP backbones for all experiments. While this choice aids interpretability, it raises questions about real-world applicability, especially given that modern DG/SSL research typically evaluates on larger, heterogeneous datasets (e.g., WILDS, DomainNet) (Kalibhat et. al. 2023) using ubiquitos ResNet/ViT-based backbones.
2. **Absence of full finetuning or transfer analysis:** The paper exclusively uses linear probe evaluation on representations learned by SCL. However, linear probes are mainly diagnostic; most real applications involve end-to-end finetuning. It remains unclear whether SCL’s benefits persist when the backbone is finetuned on downstream tasks, or whether disentanglement constraints help or hurt transfer performance.
3. **Pretraining data and SSL framing:** This setup differs from canonical SSL, which allows backbones to be pretrained on large unlabeled corpora and then transferred to labeled downstream tasks. However, pretraining here is performed on the source datasets which already have labels to leverage. With such a setup, there is no reason not to simply evaluate SOTA DG methods on the source data.
4. **No Limitations Analysis of zs-zd Disentanglement:** The assumption that semantic and domain information should be strictly independent for robust generalization is actively debated (key focus of causal-based DG methods e.g., “Causality Inspired Representation Learning” Lv et. al., 2022). For example, in many visual settings, domain-specific cues can also be semantically informative (e.g., texture or color for animal categories). The paper does not analyze such trade-offs, nor cases where zs-zd independence could suppress useful signal.

**Questions:**

1. Have the authors evaluated SCL on larger backbones (e.g., ResNet-50, ViT-B) or higher-capacity datasets (e.g., DomainNet, WILDS)? Would the disentanglement objective remain stable and beneficial at those scales?
2. How does SCL perform when the pretrained encoder is fully finetuned on a supervised task rather than linearly probed?
3. How does SCL’s effectiveness scale with fewer or more source domains? The current experiments (4 domains) are fixed; analyzing this could reveal the method’s data efficiency.
4. What is the intended pretraining scenario in practice? Could SCL be combined with a related larger unlabeled corpus (domain-labeled or not)?

---

> ### Author Response · Authors · 2025-11-19
>
> We sincerely appreciate your constructive reviews and are glad to hear that you think that our work is “well-motivated". We are glad to discuss your suggestions further.
>
> **Q1: Limited Scalability and Practicality of Experimental Setup. How does SCL’s effectiveness scale with fewer or more source domains?**
>
> **R:**
>
> Thanks for your suggestion, we have added DomainNet to our experiment. DomainNet is an extremely large dataset with six domains and approximately 0.6 million images.
>
>
> **Q2: No Limitations Analysis of zs-zd Disentanglement**
>
> **R:**
>
> We do not assume that semantic and domain factors are strictly independent in realistic settings. Instead, our method only encourages approximate independence between the learned representations $z_s$ and $z_d$ as a regularizer. Importantly, making $z_s$ domain-invariant in our formulation does not mean removing class information: the ideal objective maximizes $I(z_s; S)$ while only penalizing $I(z_s; z_d)$, and the semantic InfoNCE loss explicitly estimates $I(z_s; S)$. Thus, $z_s$ is optimized to retain as much class-discriminative signal as possible while discarding domain cues that are redundant with $z_d$.
>
> Under our data model, labels depend on the semantic factor $S$ but not on the domain factor $D$; any apparent predictive power of domain features therefore comes from spurious correlations (e.g., backgrounds or styles that co-occur with certain classes but are unstable across domains). Suppressing such $D$-related components in $z_s$ removes these shortcuts rather than true semantics. In practice, the strength of the independence constraint is controlled by $\lambda_2$: small $\lambda_2$ leaves spurious correlations in $z_s$. It harms OOD generalization, whereas overly large $\lambda_2$ starts to hurt both ID and OOD accuracy, indicating a meaningful trade-off. The best-performing regime corresponds to moderate $\lambda_2$, where $z_s$ remains highly class-discriminative while being more robust to domain shifts.

---

> > ### Author Response · Authors · 2025-11-19
> >
> > **Table X: Performance comparison on DomainNet, using each of the six domains as the target domain. Each cell reports ID accuracy / OOD accuracy (%).**
> >
> > | Method     | Clipart ID | Clipart OOD | Infograph ID | Infograph OOD | Painting ID | Painting OOD | Quickdraw ID | Quickdraw OOD | Real ID | Real OOD | Sketch ID | Sketch OOD |
> > |----------- |-----------:|------------:|-------------:|--------------:|------------:|-------------:|-------------:|--------------:|--------:|---------:|----------:|-----------:|
> > | Naive SCL  | 26.42      | 8.45        | 28.18        | 3.78          | 27.72       | 7.76         | 24.33        | 3.89          | 27.56   | 8.36     | 27.72     | 9.56       |
> > | SCL (ours) | 27.29      | 9.48        | 29.70        | 4.00          | 29.98       | 7.45         | 26.39        | 3.72          | 29.54   | 8.35     | 29.11     | 8.61       |

---

> > > ### Author Response · Authors · 2025-11-19
> > >
> > > **Q3: What is the intended pretraining scenario in practice? Could SCL be combined with a related larger unlabeled corpus (domain-labeled or not)?**
> > >
> > > **R:**
> > > Our intended use case is as a self-supervised pretraining objective for multi-domain data with lightweight domain metadata. In practice, this corresponds to settings where data are already aggregated from multiple sources (e.g., different hospitals, sensors, acquisition sites, or style domains such as “photo/sketch/cartoon”) and a coarse domain ID $d$ is available. SCL is applied at this pretraining stage using only $\{x, d\}$, and the resulting semantic representation $z_s$ is then reused for downstream tasks via linear probes or fine-tuning on both seen and unseen domains.
> > >
> > > SCL is fully compatible with additional unlabeled corpora. When the larger corpus includes domain-like metadata (dataset ID, site, camera, etc.), these can be used directly as domain labels in the domain-contrast term, thereby expanding the multi-domain pool. When such metadata are absent, the data can still be incorporated via the semantic contrast and disentanglement terms—which do not require domain labels—or via pseudo-domain labels obtained from simple clustering or heuristics. Thus, SCL is not restricted to the small benchmarks used in our experiments, but is designed to plug into standard large-scale SSL pretraining pipelines whenever a multi-domain structure (explicit or implicit) is available.

---

> > > > ### Comment · Reviewer_qLCo · 2025-11-26
> > > >
> > > > I thank the authors for engaging with the reviews; particularly providing some preliminary analysis on DomainNet and providing some more insight on how to apply SCL. However, the major weaknesses and questions (especially with respect to experimentation/analysis) still need to be addressed and for that reason I will be maintaining my original rating.

---

### Official Review · Reviewer_WRvS · 2025-11-01

**Soundness:** 2
**Presentation:** 2
**Contribution:** 2
**Rating:** 4
**Confidence:** 3

**Summary:**

This paper proposes Structured Contrastive Learning (SCL), a framework for learning disentangled semantic and domain representations from multi-domain data. The key idea is to jointly learn 1.semantic representation $z_s$ via semantic contrast, 2. domain representation $z_d$ via domain contrast, and 3. their disentanglement by minimizing mutual information $I(z_s; z_d)$. The authors provide theoretical analysis showing that minimizing $I(z_s; z_d)$ improves OOD generalization, and empirically validate SCL on MNIST-C, Rotated MNIST, PACS, and ADNI datasets.

**Strengths:**

1. Clear Problem Formulation: The paper addresses a genuine problem in multi-domain SSL - the conflation of semantic and domain factors leading to poor OOD generalization. The motivation is well-articulated.
2. The theoretical framework is mathematically sound.
3. Comprehensive Evaluation: The paper evaluates across multiple modalities (images, tabular data) and consistently shows improvements over baselines.

**Weaknesses:**

1. Limited Novelty of Core Components: The main contribution is combining InfoNCE and HSIC.
2. The proof technique is standard - combining MMD bounds with Kantorovich-Rubinstein duality offers no new insight

**Questions:**

1. Can you provide more discussions on the training dynamics that actually lead to disentanglement?
2. Please add DomainBed benchmark.

---

> ### Author Response · Authors · 2025-11-19
>
> We sincerely appreciate your efforts in reviewing our paper and providing valuable suggestions. Below, we address each concern in detail.
>
> **W1: The main contribution is combining InfoNCE and HSIC**
>
> **R:**
> We agree that InfoNCE and HSIC are standard tools. However, our main contribution is a novel information-theoretic framework for multi-domain self-supervised learning that explicitly formulates and addresses the entanglement between semantic and domain information. We introduce two latent representations, $z_s$ and $z_d$, and the ideal objective
> $$
> L = - I(z_s; S) - \lambda_1 I(z_d; D) + \lambda_2 I(z_s; z_d),
> $$
> which encodes that $z_s$ should capture semantics, $z_d$ should capture domain factors, and $I(z_s; z_d)$ should be small to avoid semantic–domain interference. To the best of our knowledge, such an explicit decomposition of self-supervised representations into semantic and domain components, together with a unified mutual-information objective, has not been explored in prior SSL or domain generalization work.
>
> Within this framework, InfoNCE and HSIC serve as concrete estimators of the mutual information terms in our two-head architecture, rather than arbitrary losses combined ad hoc. Moreover, our theory shows that the target risk bound is monotonically increasing in $I(z_s; z_d)$, so explicitly reducing this mutual information (as SCL is designed to do) tightens the generalization guarantee and provides a principled justification for our objective.
>
> ---
>
> **W2: The upper bound offers no new insight.**
>
> **R:**
> We agree that the individual analytical tools we rely on are classical. Our contribution lies in how these tools are instantiated in the SCL setting and in the resulting conclusion: the bound shows that the target-domain generalization error is explicitly controlled by the mutual information $I(z_s; z_d)$ between semantic and domain factors. In other words, our theory precisely establishes that better generalization is achieved when $I(z_s; z_d)$ is smaller, linking target performance to representation-level domain leakage. This, in turn, provides a theoretical justification for the SCL training objective, which is specifically designed to minimize $I(z_s; z_d)$ and thereby tighten the target-domain risk bound.

---

> ### Author Response · Authors · 2025-11-19
>
> **Q1: Can you provide more discussions on the training dynamics that actually lead to disentanglement?**
>
>
> **R**:
>
>  Our training objective
> $$
> L_{\mathrm{train}} = L_{\mathrm{sem}} + \lambda_1 L_{\mathrm{dom}} + \lambda_2 L_{\mathrm{sep}}
> $$
> couples three forces that shape the representations during optimization. The semantic InfoNCE term $L_{\mathrm{sem}}$ maximizes $I(z_s; z_s^+)$ by pulling together two augmented views of the same instance and pushing away other instances in the batch. In early training, the easiest way to discriminate instances is to exploit all stable factors (both semantics and domain cues), so $z_s$ initially entangles domain/style information with semantics.
>
> The domain InfoNCE term $L_{\mathrm{dom}}$, on the other hand, maximizes $I(z_d; z_d^+)$ for samples from the same domain and repels samples from different domains, explicitly encouraging $z_d$ to capture domain-specific variation. Finally, the HSIC-based separation term $L_{\mathrm{sep}}$ directly penalizes statistical dependence between $z_s$ and $z_d$ and acts as a “routing” mechanism: any component of $z_s$ that is predictable from $z_d$ (i.e., domain-correlated) receives a gradient that suppresses it in $z_s$, while $L_{\mathrm{sem}}$ simultaneously maintains the semantic signal necessary to solve the instance-discrimination task. Over training, this tug-of-war gradually reallocates domain information into $z_d$ and leaves in $z_s$ only the domain-invariant factors that are still needed to keep $L_{\mathrm{sem}}$ low. This behavior is consistent with our theoretical formulation, where maximizing $I(z_s; z_s^+)$ and $I(z_d; z_d^+)$ approximates maximizing $I(z_s; S)$ and $I(z_d; D)$, while minimizing $I(z_s; z_d)$ provably tightens the generalization bound on the target domain.
>
> Empirically, we observe signatures of this disentangling process. First, compared to the “Naive SCL” variant that disables the domain head and separation term, full SCL consistently improves both ID and OOD linear-probe accuracy; on MNIST-C, the gains are about $2.2\%$ (ID) and $2.6\%$ (OOD), and are statistically significant under both paired $t$-test and Wilcoxon signed-rank test, indicating that the separation term does more than just regularize training. Second, t-SNE visualizations show that after training, $z_d$ forms clear clusters by domain while $z_s$ no longer exhibits domain structure, suggesting that domain information has been effectively siphoned into $z_d$ and removed from $z_s$. Finally, our hyperparameter study on $(\lambda_1, \lambda_2)$ reveals a “ridge” of good performance around moderate $\lambda_2$: overly small $\lambda_2$ leads to insufficient separation (and thus more entanglement), while overly large $\lambda_2$ over-suppresses shared information and harms semantic extraction, again matching the intuition that disentanglement emerges from a balance between semantic alignment, domain alignment, and independence.
>
> **Q2: Please add the DomainBed benchmark.**
>
>
> **R:**
>
> Thanks for your advice; all of our results will run on DomainBed in later version.

---

> ### Comment · Reviewer_WRvS · 2025-11-24
> **Official Comment**
>
> I appreciate  authors' reply.
>
> After reviewing all authors' comments, I decide to keep my original score.

---

### Comment · Area_Chair_1gWe · 2025-11-19
**Start discussion**

Hi all,

The authors have submitted their response to the initial reviews, and we now enter the discussion phase.

Please review the authors' response and the comments from other reviewers. Based on the rebuttal and discussion, please update your final score if appropriate.

We welcome and encourage further discussion as needed.

Thank you for your continued contributions.

Best regards,

Your AC.

---

### Meta-Review · Area_Chair_YNwi · 2026-01-05

**Summary:**

This is a decent submission which proposes Structured Contrastive Learning (SCL), a method comprising a two-head self-supervised framework that learns (i) a semantic representation z_s​ via a semantic InfoNCE objective, (ii) a domain representation z_d​ via a domain-contrast InfoNCE objective using domain IDs, and (iii) encourages disentanglement by minimising dependence between z_s​ and z_d via an HSIC penalty. The authors also provide a theoretical bound linking reduced I(z_s;z_d) to improved target-domain risk, and evaluate mainly via linear probing on several multi-domain benchmarks.

Across the reviewers, there is agreement that the problem motivation (semantic–domain entanglement harming OOD generalisation in contrastive learning) is well articulated, and the proposed objective is clearly described.  However, the main points of disagreement (and ultimately the decisive concerns) are:

Novelty/technical contribution: Multiple reviewers view the method as largely a recombination of known building blocks (InfoNCE + HSIC/independence regularisation), with the theoretical contributions considered standard, even when correctly executed.


Empirical strength: The most negative review argues that the experiments are too small-scale, use lightweight backbones, and rely on limited benchmarks; additionally, the evaluation focuses only on linear probes, leaving uncertainty about end-to-end fine-tuning and practical downstream transfer.


Framing as SSL vs. DG: One reviewer questions whether the setup meaningfully reflects canonical large-scale SSL pretraining, since training is conducted on labelled multi-domain datasets and leverages domain labels, raising the question of why not directly compare to stronger DG pipelines or larger DG/SSL benchmarks.

Now, in rebuttal/discussion, the authors added/claimed DomainNet results and an HSIC kernel ablation, and provided a more detailed explanation of training dynamics and assumptions. These additions are helpful but (per the reviewers’ follow-ups) do not fully resolve the significant concerns about scale, downstream fine-tuning, and overall impact; the two reviewers who responded explicitly stated they would maintain their original scores after reading the rebuttal and discussion.


A further complication is that one review appears to describe a different problem setting and datasets (e.g. Office-31), which do not match the submission’s described experiments as far as I can tell.

**Reviewer Concerns:**

In terms of the concerns that were appropriately addressed:

a) Scalability (partially): The authors added DomainNet experiments (large multi-domain dataset) and reported modest improvements of SCL over a “Naive SCL” variant. This partially answers requests for a larger dataset, though it remains limited in scope (no stronger large-scale baselines/backbones).

b) Training dynamics intuition: The rebuttal provided a coherent explanation of the “tug-of-war” among semantic contrast, domain contrast, and dependence minimisation, plus qualitative/ablation evidence (e.g. t-SNE, λ_2 sensitivity).

c) HSIC kernel sensitivity: The authors added an ablation showing robustness across several kernel choices, which addresses a methodological concern.

Some less satisfactory responses:

i) Impact and novelty remain borderline: Even with the clarified framing, the method’s core still reads as a structured combination of established estimators/regularisers (InfoNCE + HSIC), and the theoretical contribution, while potentially correct, does not appear to introduce substantially new proof ideas or insights beyond emphasising a monotonic dependence on I(z_s;z_d).

ii) Insufficient “realistic” evaluation:
--  No convincing evidence that gains hold under full fine-tuning (not just linear probes).
 -- Limited exploration of modern backbones (ResNet-50/ViT) and stronger multi-domain/shift benchmarks (e.g. DomainBed was requested and not addressed; WILDS/other realistic shifts are not shown).
-- The added DomainNet results show minor absolute improvements and remain hard to interpret without stronger baselines and clearer training/compute details.

**Reviewer Scores:**

Reviewer WRvS, who scored this with a 4, explicitly stated they would keep the original score after reading the rebuttal; with fuller discussion maybe they would have been swayed but less likely.

Reviewer 2fPs, also a 4, explicitly maintained the rating after discussion; same as above.

Reviewer qLCo (score 2), and after rebuttal (including DomainNet and clarification), they still maintained the reject rating, citing that major experimental/analysis weaknesses remain; again no score change would have been expected.

Reviewer uegn, who gave this a 4, appears misaligned with the actual submission (mentions benchmarks not used here), suggesting the reviewer may have mixed submissions.

---

### Decision · Program_Chairs · 2026-01-26

Reject